# POLYPYTHIAS: STABILITY AND OUTLIERS ACROSS FIFTY LANGUAGE MODEL PRE-TRAINING RUNS

**Oskar van der Wal**[*‡]
University of Amsterdam

**Pietro Lesci**[*]
University of Cambridge

**Max Müller-Eberstein**
IT University of Copenhagen

**Naomi Saphra**
Harvard University

**Hailey Schoelkopf**[‡]
Anthropic

**Willem Zuidema**
University of Amsterdam

**Stella Biderman**[†]
EleutherAI

## ABSTRACT

The stability of language model pre-training and its effects on downstream performance are still understudied. Prior work shows that the training process can yield significantly different results in response to slight variations in initial conditions, e.g., the random seed. Crucially, the research community still lacks sufficient resources and tools to systematically investigate pre-training stability, particularly for decoder-only language models. We introduce the PolyPythias, a set of 45 new training runs for the Pythia model suite: 9 new seeds across 5 model sizes, from 14M to 410M parameters, resulting in about 7k new checkpoints that we release. Using these new 45 training runs, in addition to the 5 already available, we study the effects of different initial conditions determined by the seed—i.e., parameters' initialisation and data order—on (i) downstream performance, (ii) learned linguistic representations, and (iii) emergence of training phases. In addition to common scaling behaviours, our analyses generally reveal highly consistent training dynamics across both model sizes and initial conditions. Further, the new seeds for each model allow us to identify outlier training runs and delineate their characteristics. Our findings show the potential of using these methods to predict training stability.

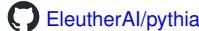 EleutherAI/pythia        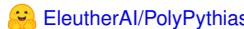 EleutherAI/PolyPythias

## 1 INTRODUCTION

Training deep learning models, including contemporary large-scale transformer-based language models (LMs), is an inherently stochastic process in which **randomness factors** (Pecher et al., 2024), such as the initialisation of model parameters and data shuffling, play a crucial role (Pham et al., 2020; Gundersen et al., 2022). Prior work on LM adaptation (Dodge et al., 2020; McCoy et al., 2020; Agarwal et al., 2021; Mosbach et al., 2021; *inter alia*) and pre-training (D'Amour et al., 2022; Madaan et al., 2024; Alzahrani et al., 2024; *inter alia*) shows that even slight variations in these randomness factors can lead to substantially different outcomes. Specifically, training multiple times using the same implementation, hardware, dataset, and hyperparameters can, nonetheless, lead to large deviations in the final performance. This variability in performance has significant implications, primarily because conclusions drawn from single training runs may be misleading or incomplete. Thus, a systematic investigation into the stability of LM pre-training is essential to ensure robustness, reproducibility, and trustworthiness in applications that use these models (Sellam et al., 2022; Goldfarb-Tarrant et al., 2024). The stability of LM performance to randomness factors in their pre-training is still underexplored, especially for recent decoder-only architectures (e.g., Radford et al., 2019). Moreover, studying learning dynamics while ensuring coverage across these randomness factors is increasingly compute-intensive due to the size of contemporary LMs and datasets.

In this work, we define **stability** as the change in a metric of interest (e.g., validation loss) caused by changes in randomness factors and quantify it using the standard deviation of that metric (see Du and Nguyen, 2023 for other approaches to quantify stability). To provide a basis for analysing the stability of LMs to randomness factors (e.g., their training dynamics or final performance) without

---

[*]Equal contribution. [†]Senior author. [‡]Work done while at EleutherAI. Correspondence to: Stella Biderman <stella@eleuther.ai>, Oskar van der Wal <oskar.vanderwal@gmail.com>, and Pietro Lesci <pl487@cam.ac.uk>.

incurring the costs to train contemporary LMs, we introduce the **PolyPythias**: an extension of the Pythia model suite (Biderman et al., 2023b) trained on the Pile dataset (Gao et al., 2021), for which we release 9 new training runs for 5 model sizes, from 14M up to 410M parameters. These new 45 training runs—in addition to the 5 already available in the suite—cover approximately 7k checkpoints across pre-training, and enable us to analyse training stability of large-scale transformer-based LM with respect to model size, parameter initialisation, and data order as quantified by metrics along the entire model training pipeline: downstream performance and consistency of predictions (§3), shifts in linguistic representations (§4), and dynamics of the model parameters and training phases (§5).

By studying the PolyPythias, we find that: (i) language modelling is largely stable and follows predictable scaling laws with respect to downstream performance; (ii) across training, we identify consistent learning phases: an initial learning phase between steps $10^3$–$10^4$ and a critical learning phase between steps $10^4$–$10^5$; (iii) using training maps constructed from statistics of the model parameters, we identify the characteristics of stable training runs and the early signals of instability.

In the following sections, we describe the PolyPythias release (§2) and how we use the multiple training runs per model size to study the stability of models across various stages of the model training pipeline (§3–§5). We conclude by combining the insights from the individual analyses (§6).

## 2 EXTENDING THE PYTHIA SUITE: POLYPYTHIAS RELEASE DESCRIPTION

The Pythia model suite (Biderman et al., 2023b)—with its open data and weights for multiple model sizes, intermediate checkpoints, and detailed reporting of the training configurations—allows researchers to study the learning dynamics of realistic LMs without the need to train them from scratch. Since its release, the suite has been extensively used to study, e.g., LMs' learning dynamics (Michaelov and Bergen, 2023; Arnold et al., 2024), memorisation patterns (Biderman et al., 2023a; Lesci et al., 2024), and biases (Hu et al., 2024). The suite is composed of 10 models ranging in size from 14M to 12B parameters trained on the Pile dataset (Gao et al., 2021; Biderman et al., 2022), a 300B-token curated collection of English documents.[1] All models are trained using the same data. Specifically, the dataset is shuffled and "packed" into sequences of $2,049$[2] tokens. Training was performed using a cosine learning rate schedule with warm-up, and using a batch size of 1,024 sequences, resulting in exactly 143k optimisation steps. In the original Pythia suite, for each model size, a single training run is available and consists of 154 checkpoints: at initialisation (step 0), log-spaced up to step 1k (steps $1, 2, ..., 512$), and every 1k steps afterwards (steps 1k–143k).

We consider models with 14M, 31M, 70M, 160M, and 410M parameters. For each size, we release 9 additional training runs resulting in about 7k new checkpoints. In App. E, Table 5, we report the links to the model checkpoints. Each training run uses the same hyperparameters, codebase, and data as Biderman et al. (2023b) but varies the seeds for parameter initialisation and batch composition.[3] We use the standard (i.e., non-deduplicated) version of the Pile and release the tokenised and pre-shuffled datasets corresponding to the different seeds. More training details are in App. A.

A limitation of our suite is that it spans model sizes up to 410M parameters. This choice reflects computational constraints, prioritising seed exploration and checkpoint granularity over scaling up model size. Our aim is to provide an additional resource for researchers unable to train even 410M parameter models from scratch, thus enabling them to study training stability across model sizes.

Prior work that released multi-seed model suites includes Sellam et al. (2022) who introduced the MultiBERTs, a set of 25 BERT-base (final) checkpoints trained with similar hyper-parameters to the original encoder-only BERT architecture (Devlin et al., 2019) but with different random seeds. However, only 28 intermediate checkpoints are available for 5 of the runs, and the release is limited to encoder-only models in a single size. Karamcheti et al. (2021) introduced 10 GPT-2 (124M and 355M parameters; Radford et al., 2019) training runs, each with 600 intermediate checkpoints, but still limited to two model sizes. More recently, Madaan et al. (2024) trained a suite of 10 Llama-2-7B

---

[1]There exists a deduplicated version of the Pile dataset used to train a second version of the Pythia suite.

[2]Target tokens are the right-shifted input tokens; thus, an additional token is required to achieve the desired input and target sequence length of 2,048 tokens.

[3]We will use the term "batch composition" instead of "data order" because the GPT-NeoX codebase (Andonian et al., 2023), used to train the (Poly)Pythias, shuffles documents before packing them into sequences. This results in sequences that are not simply reshuffled across seeds; they are unique due to the different packing.

(Touvron et al., 2023) models initialised with different random seeds on 210B tokens, analysing 21 intermediate checkpoints which, however, remain publicly unavailable. PolyPythia compares favourably to these suites by spanning 5 model sizes with 154 checkpoints per run, using 10 seeds per model, for a total of almost 7k checkpoints trained on publicly available data.

## 3 STABILITY OF DOWNSTREAM PERFORMANCE

> We start by asking a key question, especially relevant for practitioners: *"How stable is model performance on downstream tasks to randomness factors?"* We first study how performance varies across seeds (controlling both batch composition and parameters' initialisation) for a given model size. Then, we analyse how models' predictions and learned gender biases vary throughout the training process and across seeds. We find that language modelling is largely stable and follows predictable scaling laws with respect to downstream performance.

Following Biderman et al. (2023b), we measure model performance as the average **accuracy** on a set of multiple-choice tasks: ARC (Easy) and ARC (Challenge) (Clark et al., 2018), LAMBADA (Paperno et al., 2016), Logiqa (Liu et al., 2020), Piqa (Bisk et al., 2020), SciQ (Welbl et al., 2017), WinoGrande (Sakaguchi et al., 2020), and WSC (Levesque et al., 2012). We measure how predictions agree across seeds and throughout training by computing the Cohen's $\kappa$ (Cohen, 1960) on the individual multiple-choice answers, where $\kappa = 1$ means perfect agreement while $\kappa = 0$ denotes agreement at the chance level. Specifically, we define the **inter-seed agreement** as the Cohen's $\kappa$ between the predictions of a model trained with a particular seed and the same model trained with seed 0. Additionally, we define **self-consistency** as the Cohen's $\kappa$ between the predictions of a model at the last checkpoint and any previous one. Finally, we measure a model's gender bias as its accuracy on the BLiMP (Gender Agreement) (Warstadt et al., 2020), CrowS-Pairs (Gender) (Nangia et al., 2020), and Simple Co-occurrence Bias (Smith et al., 2022) benchmarks. More details about these benchmarks in App. B.

We perform the evaluation using the Language Model Evaluation Harness framework[4] (Gao et al., 2024; Biderman et al., 2024). To limit computational costs while being able to track model behaviour across training for each size and seed, we evaluate performance on a subset of the available checkpoints. Specifically, we use checkpoints at (log-spaced) steps $0, 1, 2, ..., 512, 1\text{k}$, and from step $3\text{k}$ onwards we choose every $10\text{k}$-th step up to $143\text{k}$, the final checkpoint. We report accuracy, inter-seed agreement, and self-consistency on ARC (Easy) and SciQ in Fig. 1 and the gender bias results in Fig. 2. We show the other benchmarks in App. D. For each metric, we show the median and interquartile range across seeds. We discuss the individual results below.

**Downstream Performance.** We find (unsurprisingly) that the larger models consistently outperform their smaller counterparts, as indicated by higher accuracy (Fig. 1, left column), except for the more challenging tasks for which all models perform as good as random (see ARC (Challenge), Logiqa, WSC, WinoGrande in App. D, Fig. 6), a result consistent with Biderman et al., 2023b. Performance improves most after step $10^3$. However, it drops between step $10^4$ and $10^5$ for all but 410M, especially for the smallest models. This finding aligns with prior work showing that smaller LMs suffer from "saturation", i.e., a drop in performance at a later stage of the training process due to a mismatch between the dimension of the model representations and the high rank of the output embedding matrix (Michaelov and Bergen, 2023; Godey et al., 2024).

**Inter-Seed Agreement and Self-Consistency.** For a more fine-grained analysis of how much model behaviour changes across seeds, we compare inter-seed agreement and self-consistency in Fig. 1 (middle and right column, respectively). Inter-seed agreement peaks at $\kappa \approx 0.7$ around step $10^3$ before converging towards a "moderate agreement"—i.e., $\kappa \approx 0.5$—around step $10^4$ and remaining at this level until the end of training. Self-consistency steadily increases to "moderate agreement"—i.e., $\kappa \approx 0.5$—up to step $10^3$, after which it plateaus up to step $10^4$, before continuing to increase until the end of training as the models settle on their final answers. This finding aligns with the slow convergence observed in small models by Diehl Martinez et al. (2024).

**What is special about step $10^3$?** A qualitative analysis of model predictions, across tasks and seeds, shows that around step $10^3$ models begin generating non-random answers and accuracy

---
[4] github.com/EleutherAI/lm-evaluation-harness.

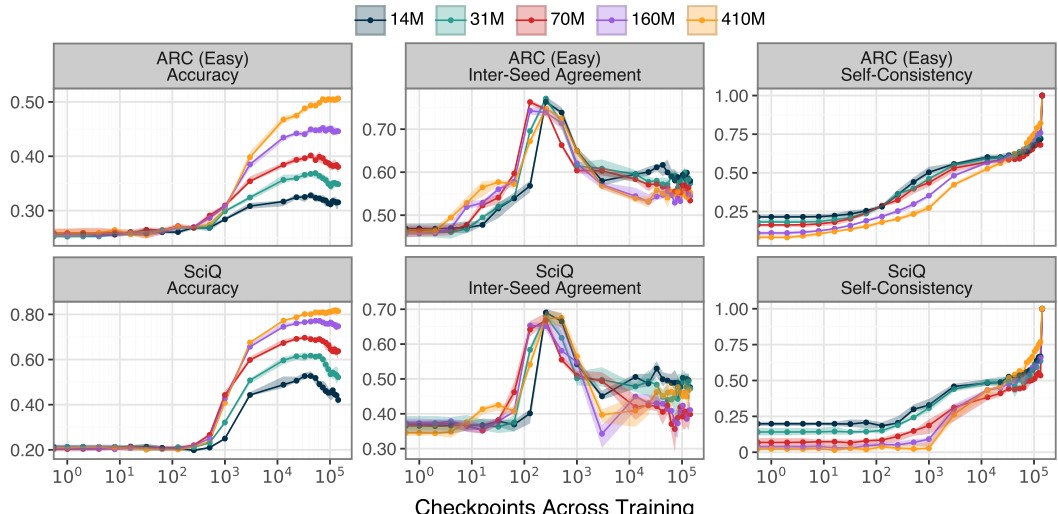

Figure 1: Accuracy, Inter-Seed Agreement, and Self-Consistency (median and interquartile range across seeds) on ARC (Easy) and SciQ.

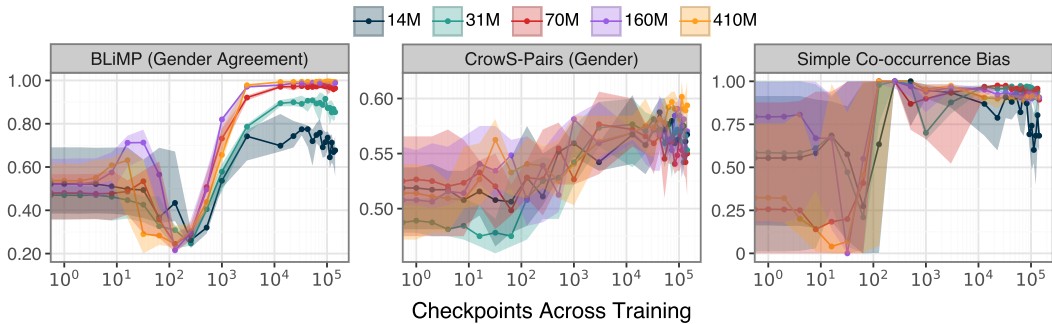

Figure 2: Accuracy, proportions of times the stereotypical answer is chosen, and proportion of times the male option is preferred for, respectively, BLiMP (Gender Agreement), CrowS-Pairs (Gender), Simple Co-occurrence Bias (median and interquartile range across seeds).

improves. Since PolyPythias use a batch size of slightly over 2M tokens, this step aligns with the 2.5B-token mark identified by Olsson et al. (2022) as the point where induction heads form and in-context learning capabilities emerge. Also, this aligns with Tigges et al. (2024) who observe a variety of circuits emerge in Pythia models, regardless of model size, between 2B and 10B tokens (i.e., steps 1k –5k)—e.g., induction heads, successor heads (Gould et al., 2024), copy-suppression heads (McDougall et al., 2023), and name-mover heads (Wang et al., 2023). Further, Chang and Bergen (2022) find that early in training LMs primarily rely on unigram token frequencies before gradually shifting to more contextual predictions—a finding further corroborated by Meister et al. (2023). Similarly, Jumelet et al. (2024) show that Pythia models develop adjective order preferences within this same training range. Collectively, these findings suggest that core semantic functions emerge at a consistent stage in training, regardless of model size or randomness factors.

**Finding outlier seeds based on Accuracy and Validation Loss.** Zooming in on Fig. 1, we see that per-size accuracy is generally consistent across seeds, with two exceptions. For a given model size, we can identify outlier seeds as follows.[5] First, we consider the accuracy of the last checkpoints on the ARC (Easy), LAMBADA, Piqa, and SciQ tasks, for which all models perform better than random. Second, for each task and model size, we standardise accuracy to have mean zero and a standard deviation of one by standardising across seeds. Finally, we define a region of 2 standard deviations from the mean of a model on that task and consider "outliers" those model-seed combinations that fall outside this region. We (remarkably) find only two such combinations: 410M seed 3 and 4; only

---

[5]A formal statistical test (e.g., ANOVA and Tukey's test) would have required a larger sample size (i.e., more tasks). Nonetheless, this simple heuristic allows us to discover the same outliers we find using other approaches.

for these very seeds we observe "loss spikes" (see Fig. 5 in App. A). We will further explore these outlier seeds using other metrics and approaches in §5.

**Gender Bias.** We find distinct phases in the development of gender, both grammatical and bias, in Fig. 2. Specifically, the models start to learn gender agreement between step $10^2$ and $10^3$ (see BLiMP (Gender Agreement)), which coincides with a sudden shift to a strong bias for using "male identifier words" (see Simple Co-occurrence Bias). Around this step, but less sharply, we also see an increase in gender bias measured by the CrowS-Pairs (Gender) benchmark, which measures more semantically diverse stereotypes than simple co-occurrence statistics. We posit that the large variance observed for the bias measures reflects the poor reliability (e.g., due to small benchmark size, poor quality test items, etc) rather than actual bias differences (see Van der Wal et al., 2024; Delobelle et al., 2024).

## 4 REPRESENTATIONAL STABILITY OF LINGUISTIC INFORMATION

> We now focus on the step before output generation and analyse token representations and find that they remain similar across seeds. Also, representational stability follows consistent trajectories across model sizes, suggesting that trends in smaller models reliably predict those in larger ones.

We study representational (in)stability by applying the information-theoretic probing approach proposed in Müller-Eberstein et al. (2023).[6] We consider seven linguistically motivated token-classification tasks: coreference resolution (Coref; Pradhan et al., 2013), dependency parsing (Dep; Silveira et al., 2014), named entity recognition (NER; Pradhan et al., 2013), part-of-speech tagging (PoS; Pradhan et al., 2013), semantic tagging (SemTag; Abzianidze et al., 2017), sentiment analysis (Senti; Socher et al., 2013), and topic classification (Topic; Lang, 1995); more details in App. B. For each task, we train a **probe** classifier $\boldsymbol{\theta} \in \mathbb{R}^{d \times |\mathcal{Y}|}$ as follows. First, given an input-output pair, $(\boldsymbol{x}, \boldsymbol{y}) \in \mathcal{D}$, we collect the model representations, $\mathbf{h}_l(x) \in \mathbb{R}^d$, for each token $x$ in the input sequence $\boldsymbol{x}$ at each layer $l$. Then, we aggregate per-layer representations into a global representation for that token using a learned weighting scheme $\boldsymbol{\alpha} \in \mathbb{R}^L$, obtaining $\mathbf{h}(x) = \sum_{l=1}^{L} \alpha_l \mathbf{h}_l(x)$. This global representation is finally passed as input to the probe, which outputs class probabilities for that token. Both $\boldsymbol{\alpha}$ and $\boldsymbol{\theta}$ are jointly learned by optimising the minimum description length (MDL) loss:[7]

$$\mathcal{L}_{\text{MDL}} = - \mathbb{E}_{\boldsymbol{\theta} \sim p(\boldsymbol{\theta})} \left[ \sum_{(\boldsymbol{x}, \boldsymbol{y}) \in \mathcal{D}} \underbrace{\left( \sum_{x \in \boldsymbol{x}, y \in \boldsymbol{y}} \overbrace{\mathcal{L}_{\text{CE}} \left( \boldsymbol{\theta}^\top \mathbf{h}(x), y \right)}^{\text{Token-level loss}} \right)}_{\text{Sequence-level loss}} \right] + \text{KL}\big( p(\boldsymbol{\theta}) \, \| \, q(\boldsymbol{\theta}) \big) \quad (1)$$

where $\mathcal{L}_{\text{CE}}$ is the cross-entropy loss. Probes trained with only cross-entropy loss can correctly map random representations to labels (Voita and Titov, 2020; Pimentel et al., 2020). Thus, eq. (1) includes a KL-divergence term to keep the probe's distribution $p(\boldsymbol{\theta})$ close to a sparsity-inducing prior $q(\boldsymbol{\theta})$.

We study representational stability through three metrics. First, we measure the **information content** of representations using the probe's macro-F1 score. Second, we note that eq. (1) corresponds to the probe's codelength; thus, we measure **representational efficiency** by defining the **codelength ratio** between random[8] *vs.* probe's representations where values close to 0 indicate high representational efficiency. Finally, we measure the **representational shift** between two consecutive checkpoints using the principal **subspace angles** (**SSA**s; Knyazev and Argentati, 2002): given two probes, SSAs return an angle between $0°$ and $90°$ where lower values represent more similar representations.

Results are summarised in Fig. 3, which shows the macro-F1, Codelength Ratio, and SSAs (rows) for each task (colour) averaged across seeds (line) and the respective standard deviation (shaded area) for each model size (columns) and checkpoint (x-axis). We discuss these results and explore their correlation across different model sizes below.

**Information Content.** The macro-F1 scores mirror (unsurprisingly) the scaling behaviour observed for the ARC (Easy) and SciQ tasks (§3), where scores consistently improve with model size across

---

[6]For space reasons, we only sketch the method here and refer to the original paper for a detailed introduction.
[7]We refer to Voita and Titov (2020) for a formal derivation.
[8]We use the probes obtained from the randomly initialized models at step 0.

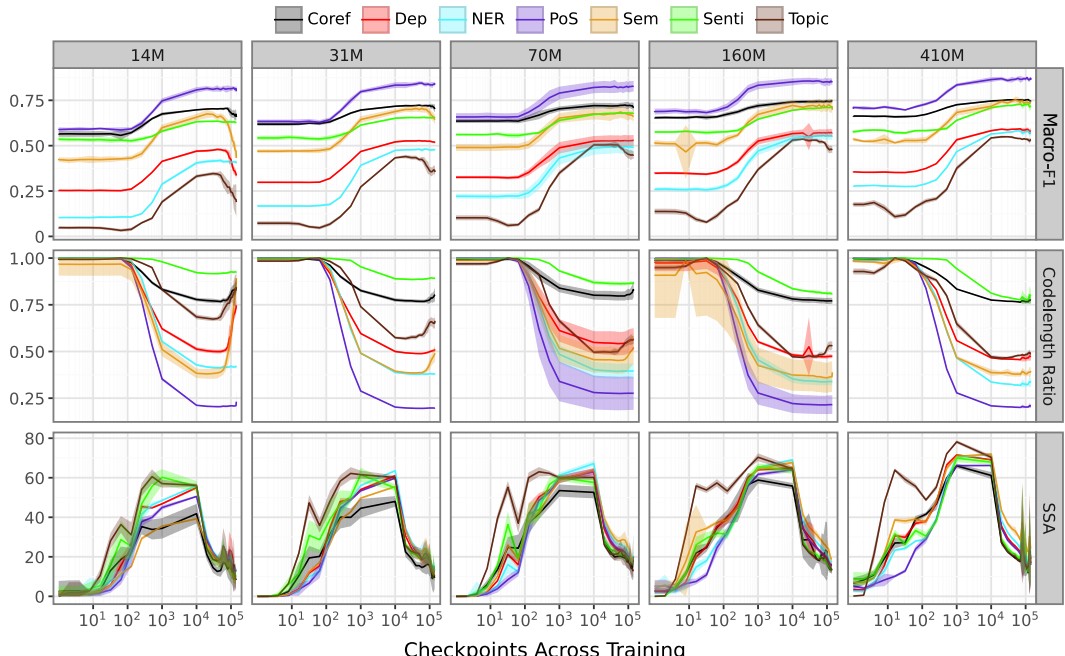

Figure 3: Changes in latent representations of linguistic phenomena (coreference, syntactic dependencies, named entities, parts-of-speech, semantic tags, sentiment, topic). macro-F1, Codelength Ratio, and SSA (rows) for each linguistic task (colour) averaged across seeds (line) and the respective standard deviation (shaded area) for each model size (columns) and checkpoint (x-axis).

both lower-level syntactic tasks (e.g., PoS, Dep) and higher-level semantic ones (e.g., NER, Senti). In other words, larger models return representations with higher information content. While final values differ across model sizes, their trajectory is consistent: mirroring accuracy scores in Fig. 1 (top row), we observe initial improvements around step $10^3$ and a rapid increase in macro-F1 up to step $10^4$, after which performance for most tasks remains stable, except for the smaller model sizes.

**Representational Efficiency.** Like macro-F1, the Codelength Ratio starts improving around step $10^3$ and reaches its stable value around step $10^4$. The final Codelength Ratio for each task depends on the model size, with a distinction between models below and above 100M parameters. Specifically, we observe similar final values for syntactic tasks (e.g., PoS, Dep, Coref) across model sizes, while larger models achieve a 5-10% abs. lower Codelength Ratio on semantic tasks (e.g., NER, Senti).

**Representational Shift.** The SSAs follow (remarkably) similar trajectories across model sizes. For each task and model size, SSA increases up to step $10^3$, then plateaus until step $10^4$, and finally decreases until the end of training to a value around $20°$. In other words, model representations after step $10^4$ tend to become more stable, matching the reduced rate of change in macro-F1 and Codelength Ratio. The SSA does not directly correlate with macro-F1 or Codelength Ratio: larger changes in SSA early on have positive effects, while smaller changes later on can negatively affect these metrics (e.g., 14M). However, we note that SSA roughly follows the learning rate schedule: peaking around step 2k (end of the warm-up phase) and then decreasing until the end of training.

**Correlation across model sizes.** We investigate the similarity of the trajectories of representational stability metrics across model sizes. A metric's trajectory for a model size is based on the concatenation of its values for all tasks across all checkpoints. We first compute the Pearson correlation $r_{i,j}$ between a metric's trajectory for each pair of model sizes $\langle i, j \rangle \in \mathcal{M} \times \mathcal{M}$ where $\mathcal{M} = \{14\text{M}, 31\text{M}, 70\text{M}, 160\text{M}, 410\text{M}\}$. To obtain the average $\bar{r}$ across all pairs of model sizes, we use the Fisher transformation over correlation coefficients (Fisher, 1970): $\bar{r} = \tanh\left(\frac{1}{|\mathcal{M}|^2} \sum_{i,j \in \mathcal{M} \times \mathcal{M}} \operatorname{arctanh}(r_{i,j})\right)$. Finally, we average the resulting $\bar{r}$'s across seeds in the same way to obtain per-metric average correlations. We find that macro-F1, Codelength Ratio, and SSA are highly correlated across model sizes with an average $\bar{r}$ of 0.99 for macro-F1, 0.98 for

Codelength Ratio, and $0.94$ for SSA (all with a *p*-value smaller than $0.001$). This indicates that the representational stability of smaller models is strongly indicative of that of larger models.

## 5 TRAINING PHASES AND OUTLIER SEEDS

> After analysing the stability of performance and intermediate representations, we now examine the dynamics of model parameters using training maps. We find mostly consistent training dynamics across model sizes and seeds, with some exceptions. Additionally, training maps from smaller models can predict those of larger models and their final performance.

We investigate the dynamics of model parameters using training maps. A **training map** (Hu et al., 2023) associates each checkpoint with a latent state by fitting a Hidden Markov Model (HMM; Baum and Petrie, 1966) to a vector of statistics derived from the model parameters (e.g., $L_2$-norm). To fit the HMM, we first gather statistics—listed in App. C, Table 4—from all checkpoints. For each checkpoint within each model size, we standardise the statistics across seeds so that they have a mean of zero and a standard deviation of one, as HMMs are sensitive to the scale of the inputs. We use the standardised sequence to train the HMM with the Baum-Welch algorithm (Baum et al., 1970). Typically, the number of latent states, the primary hyperparameter in HMMs, is determined by minimising some information criterion (e.g., Schwarz, 1978; Akaike, 1998) computed on a validation set. To enable comparisons across model sizes, we use $5$ states, as this value is near optimal for the Bayesian Information Criterion across all sizes. Consequently, for each training run, we obtain a sequence of latent states (one per each checkpoint) representing its training map.

First, we use training maps to compare training runs and find outliers (Fig. 4), and we report which properties of the parameters drive state transitions (Table 1). Second, we study the relationship between training maps and final model performance (Tables 2 and 3), and investigate whether it is possible to zero-shot predict final performance from the training map alone. Specifically, we represent each training map as a **bag-of-states**, i.e., a vector counting the number of times a model visits each state. We then use this as input for a linear regression model to predict final performance. As our performance metric, we use the average accuracy of the final model checkpoint on ARC (Easy), LAMBADA, Piqa, and SciQ; we choose these tasks as all models perform better than random. To average accuracy across tasks consistently, for each model size, we standardise it across seeds to have a mean of zero and a standard deviation of one. We refer to the resulting scores as **z-scores**.

**Characterising the training maps of outlier seeds.** Across model sizes and seeds, we find that training maps are linear graphs with a few exceptions for specific seeds, i.e., seed 3 and 4 of model size 410M. In Fig. 4, we visualise the HMM (left) and training map (right) for 410M and show outlier (top) and stable (bottom) seeds separately. In line with Hu et al. (2023), we find that linear training maps (Fig. 4, bottom-left) describe stable dynamics and performance (Fig. 4, bottom-right), while maps with "forks"—i.e., regressions to an earlier state—are associated with instability. Specifically, forks are only present in the training maps of the outlier seeds (Fig. 4, top-left); these seeds showcase sudden drops in performance (Fig. 4, top-right) and loss spikes (see Fig. 5 in App. A).

**Drivers of state transition.** In Table 1, we report the three main drivers of state transition, focusing on the transitions that appear in the stable maps but not in the outlier maps and those that are only present in the outlier maps. First, outlier maps fail to perform the transition ②→③ in which the parameters $L_2$, median bias, and the average weight variance decrease. Abnormal state transitions for the outlier maps (②→④) are driven by an increase in the variance of the weights' singular values ($\sigma_\lambda$). The subsequent state transitions (④→⓪ and ④→①) coincide with a strong performance drop and are driven by a sharp decrease in $\sigma_\lambda$. This phenomenon is also described by Godey et al. (2024) as "representation degeneration": the distribution of singular values first becomes increasingly uniform and then abruptly degenerates around a point. We leave further exploration as future work.

**Consistency in state transitions.** In Table 2 (right), we report the steps at which state transitions occur. We find that state transitions mostly happen around the same step for each model size and across different seeds. The only exception is 410M, for which we observe variability caused by outlier seeds. Additionally, transitions tend to occur at similar steps: the first transition happens around step $10^3$, followed by two more transitions between $10^3$ and $10^4$, and the final transition

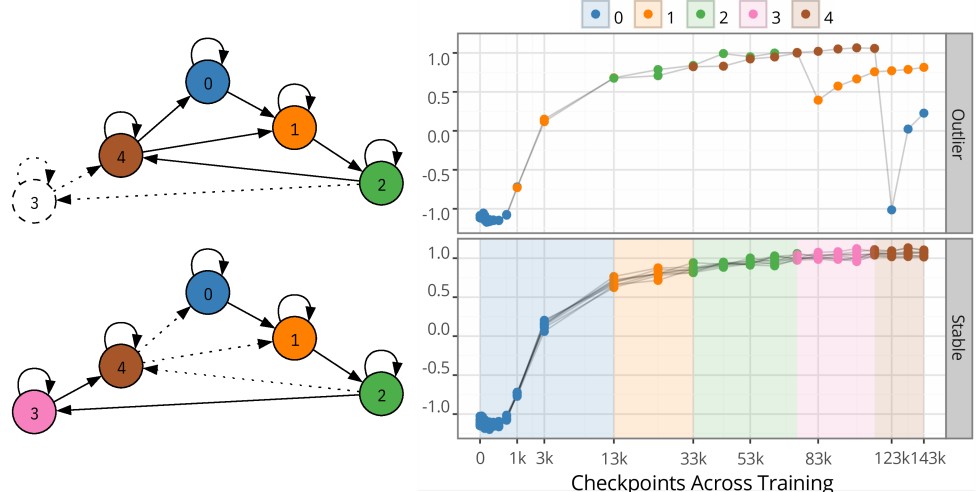

Figure 4: Average standardised accuracy (z-score) across the datasets ARC (Easy), LAMBADA, Piqa, and SciQ (right) for each seed of the model size 410M, along with their corresponding training maps (left). The HMM transitions are colour-coded according to the training map. The results are divided into outlier runs (top; seeds 3 and 4) and stable runs (bottom).

| Transition | Description | Top 3 feature determining each transition | | |
|---|---|---|---|---|
| ② → ③ | Missing in Outlier | $\text{median}_b \downarrow 0.44$ | $L_2 \downarrow 0.42$ | $\sigma_w \downarrow 0.50$ |
| ② → ④ | Only in Outlier | $\lambda_{\max} \uparrow 1.41$ | $\sigma_b \uparrow 1.79$ | $\sigma_\lambda \uparrow 1.71$ |
| ④ → ⓪ | Only in Outlier | $L_1/L_2 \uparrow 2.76$ | $\sigma_b \downarrow 2.43$ | $\sigma_\lambda \downarrow 2.99$ |
| ④ → ① | Only in Outlier | $L_1/L_2 \uparrow 2.20$ | $\sigma_\lambda \downarrow 2.21$ | $\lambda_{\max} \downarrow 1.94$ |

Table 1: Top three drivers of state transitions for 410M. Focus on transitions unique to stable maps and those only present in outlier maps.

around $10^5$. Notably, 410M is less consistent, even when we do not account for the outlier seeds. We leave the exploration of this aspect for future work.

**Training maps and final performance.** The training maps (interestingly) reveal that the outlier runs start to deviate from the other runs long before they show worse scores on the performance metrics. In Fig. 4 (top), we see that both runs enter state ② prematurely and fail to make the transition to state ③. To investigate whether a model's training maps are informative of that model's final performance, we perform linear regressions using the bag-of-states to predict the final average z-score. We report the regression $R^2$ (Table 2, $R^2$ column). Only the training map of 410M is predictive of performance ($R^2 = 0.99$). Also, this model, due to the outlier seeds, is associated with a high performance variance (Table 2, $\sigma^2$ column). We report the regression coefficients for 410M in the last row of Table 3. We observe that the outlier seeds (3 and 4) are the only ones receiving a negative coefficient. In other words, the bag-of-states obtained from the training map of 410M has enough information to predict this model will underperform.

**Predicting performance across model sizes using training maps.** We investigate whether training maps from smaller models can help predict the performance of a larger model. Specifically, we attempt to predict the performance of 410M using training maps from smaller models. Our approach consists of two steps. First, for a given small model (e.g., 14M), we train an HMM on checkpoints from all seeds. We then use this HMM to assign each checkpoint of 410M to a latent state, effectively performing a "zero-shot" prediction of its training map. Second, we aggregate the training maps of the large model across all seeds into a bag-of-states representation, which we use to predict the performance of each seed. Our hypothesis is that if this regression successfully predicts model performance, then the training map of the smaller model carries meaningful information that can be used to predict the performance of the larger model. In Table 3, we report the $R^2$ and coefficients of the regression that uses each size-specific training map to predict the performance of 410M. We find

| | | z-scores | | Step at which transition happens[*] | | | |
|---|---|---|---|---|---|---|---|
| Size | Fork | $R^2$ | $\sigma^2$ | (0) → (1) | (1) → (2) | (2) → (3) | (3) → (4) |
| 14M | | 0.75 | 0.65 | $5k_{\pm 0.7k}$ | $38k_{\pm 3.5k}$ | $72k_{\pm 2.2k}$ | $108k_{\pm 2.7k}$ |
| 31M | | 0.64 | 0.84 | $7k_{\pm 0.7k}$ | $42k_{\pm 2.6k}$ | $69k_{\pm 2.1k}$ | $104k_{\pm 2.0k}$ |
| 70M | | 0.41 | 0.71 | $6k_{\pm 0.6k}$ | $28k_{\pm 3.1k}$ | $58k_{\pm 3.3k}$ | $94k_{\pm 2.5k}$ |
| 160M | | 0.03 | 0.58 | $2k_{\pm 0}$ | $18k_{\pm 0.8k}$ | $61k_{\pm 1.5k}$ | $100k_{\pm 1.6k}$ |
| 410M | ✓ | 0.99 | 0.98 | $18k_{\pm 22.6k}$ | $35k_{\pm 24.1k}$ | $73k_{\pm 20.9k}$ | $114k_{\pm 4.9k}$ |

Table 2: Overview of statistics for the training maps found for the different sizes of Pythia models. $R^2$: the goodness of fit for the linear regression for the bag-of-states and the z-scores. $\sigma^2$: the variance of the average performance z-scores. [*]Averaged across seeds, except for outlier seeds 3 and 4 for 410M to remove non-linearities due to forks in the training maps.

| | | Seed used to train 410M | | | | | | | | | |
|---|---|---|---|---|---|---|---|---|---|---|---|
| HMM | $R^2$ | 0 | 1 | 2 | 3 | 4 | 5 | 6 | 7 | 8 | 9 |
| 14M | 0.18 | 0.01 | 0.06 | 0.39 | −1.12 | −0.21 | 0.03 | 0.35 | 0.01 | 0.08 | 0.39 |
| 31M | 0.97 | 0.42 | 0.52 | 0.48 | −0.56 | −2.78 | 0.42 | 0.31 | 0.32 | 0.44 | 0.43 |
| 70M | 0.97 | 0.34 | 0.43 | 0.48 | −0.60 | −2.75 | 0.23 | 0.19 | 0.51 | 0.67 | 0.49 |
| 160M | 0.96 | 0.46 | 0.46 | 0.69 | −0.58 | −2.74 | 0.31 | 0.28 | 0.37 | 0.25 | 0.50 |
| 410M | 0.99 | 0.39 | 0.45 | 0.38 | −0.57 | −2.80 | 0.47 | 0.48 | 0.42 | 0.49 | 0.29 |

Table 3: Linear regression coefficients returned by regressing the average z-score of 410M on its bag-of-states obtained from the zero-shot training map constructed using the HMM trained on the model listed in column "HMM". Outlier seeds are highlighted in  grey .

that forming the bag-of-states using predict the average z-score successfully with $R^2 > 0.9$ for all models, except 14M, the smallest size. Also, the negative coefficients for the outlier seeds indicate that it is possible to predict underperforming runs across sizes. When computing the bag-of-states for 410M, we pass all its checkpoints through the HMM to generate the full training map. Ideally, we would like to use only the initial checkpoints to predict the final performance of a partially trained model. This would allow us to decide early on whether to stop a specific run. However, when we construct the bag-of-states using only a partial training run, we fail to predict the average z-score accurately. Empirically, we find that at least 120k steps (out of the total 143k) are required for a reliable prediction. We leave the investigation of which properties can be predicted from the bag-of-states of early checkpoint metrics as future work.

## 6 DISCUSSION

Our experiments on downstream performance (§3), intermediate representations (§4), and model parameters (§5) allow us to examine the stability of training and find outlier runs using different methods across the model training pipeline. In this section, we analyse commonalities across the resulting metrics to identify broader characteristics of LM pre-training dynamics.

**Language modelling is largely stable.** Generally, we observe LM pre-training dynamics to follow consistent trajectories. Across seeds and model sizes, downstream performance and representational efficiency consistently increase during pre-training, and training maps are linear (except for the outlier seeds). Furthermore, model scaling laws seem to hold across seeds, not only for downstream performance but also for information content and representational efficiency of model representations. Similarly, both at the performance and representational level, we observe the effect of "saturation" (Michaelov and Bergen, 2023; Godey et al., 2024) in smaller models.

**Linguistic information is encoded in the initial learning phase ($10^3$–$10^4$ steps).** Across all experiments, metrics start moving away from the initial random baseline around step $10^3$ (2B tokens circa) and reach their convergence level around step $10^4$ (20B tokens circa). In this initial phase, representational shift peaks and linguistic information begins to be encoded into the models' latent representations. Through the lens of multiple metrics, we can analyse model behaviour in this phase

in detail. Specifically, while the amount and the efficiency with which linguistic information is encoded in model representations have already increased substantially at step $10^3$, the model does not yet generate coherent outputs, as indicated by low performance on linguistic acceptability benchmarks like BLiMP (Gender Agreement). Simultaneously, self-consistency is low while inter-seed agreement is high, which we hypothesise to be an artefact of all models initially choosing an incorrect baseline answer. In terms of training maps, this phase corresponds to the (0) → (1) transition, which occurs consistently in this initial training phase for all model sizes (except for the 410M outliers).

**Most improvements happen in the "critical" learning phase ($10^4$–$10^5$ steps).** In the range of $10^3$ to $10^4$ steps, most learning occurs, as measured by all of our metrics. Performance increases the most, and linguistic information content and representational efficiency converge to close-to-final values. At the same time, the representational shift begins to decrease from its peak, showing how the information encoded in the model begins to stabilise. This is reflected in terms of performance by the simultaneous increase in self-consistency. The fact that inter-seed agreement decreases before remaining relatively constant until the end of training indicates that models settle on their final answers after this stage. In the training maps, this phase corresponds to the (1) → (2) and (2) → (3) transitions that occur before step $10^5$. The fact that our outlier runs (410M, seeds 3 and 4) exit state (2) prematurely indicates that this phase is important for the LM's downstream performance. Furthermore, we note that these learning phases are (remarkably) similar and occur at the same time during pre-training as for encoder-only models (Müller-Eberstein et al., 2023). Also, they follow similar trajectories as recurrent LSTM architectures (Saphra and Lopez, 2019). We leave the exploration of which modelling or data decisions may result in these learning phases for future work. Finally, we observe that all benchmarks still improve past the optimal token count (e.g., 8.2B tokens, or around 4k–5k steps) predicted by the Chinchilla scaling law (Hoffmann et al., 2022).

**Training maps describe outlier seeds.** Using the downstream performance results and training loss (§3), and training maps (§5), we identify outlier seeds and explain how the model parameters change in the unstable pre-training regime. Moreover, we find that it is possible to predict training maps in a "zero-shot" fashion (e.g., using HMM trained on smaller models to predict the training map of larger ones), suggesting that statistics of the parameters of smaller models are informative of their larger counterparts. We propose using PolyPythias to investigate potential connections between the state transitions observed in training maps and the emergence of components or behaviours in circuit analyses, as discussed in §3. However, a limitation is that early-training state transitions may occur too consistently across seeds for each model size to provide meaningful insights into this relationship. To better understand whether connections exist between circuit formation (e.g., the development of induction heads or other circuit components in each seed) and state transitions, interventional studies of training dynamics may be necessary. We leave the exploration of this aspect for future work.

## 7    CONCLUSIONS

In this work, we introduce PolyPythia, a multi-seed extension of the Pythia model suite (Biderman et al., 2023b), adding 45 extra pre-training runs for a total of 10 seeds across 5 model sizes. This expanded resource is designed to facilitate research on training dynamics and model stability. Through our experiments, we demonstrate the usefulness of PolyPythias by analysing the stability of the language modelling pipeline. We examine downstream performance, intermediate representations, and parameter training dynamics across seeds and model scales. Our findings suggest that, with some exceptions, language modelling remains largely stable.

We hope these additional seeds will support further research into the impact of randomness in model pre-training. This includes studying the robustness of different evaluation metrics (e.g., Sellam et al., 2022; Van der Wal et al., 2024; Madaan et al., 2024), identifying factors contributing to suboptimal training (e.g., Zoph et al., 2022; Zhai et al., 2023; Chowdhery et al., 2023; Zeng et al., 2023; Chung et al., 2024), and improving the predictability of model performance across scales (e.g., Kaplan et al., 2020; Srivastava et al., 2023). Further, the diverse training runs with different data orders enable further studies on memorisation (e.g., Biderman et al., 2023a; Lesci et al., 2024) or on the relationship of the data to the emergence of learned behaviours (Van der Wal et al., 2022; Biderman et al., 2023b; Jumelet et al., 2024; Belrose et al., 2024). Lastly, PolyPythia provides a valuable testbed for assessing benchmark evaluations' reliability and model interventions' effectiveness.

## ACKNOWLEDGEMENTS

This research was made possible through computational resources generously provided by StabilityAI. We express our gratitude to Michael Hu for his assistance with the implementation of the HMM training maps presented in §5. We thank Andreas Vlachos, Tiago Pimentel, and Clara Meister for their insightful feedback on earlier drafts. Finally, we extend our thanks to Davide Lesci and Marco Lesci for proofreading the final version of the manuscript.

**Oskar van der Wal** initiated this project during his research internship at EleutherAI and gratefully acknowledges their support. His work is partially funded by the Dutch Research Council (NWO) through the project "The biased reality of online media" (406.DI.19.059). The views expressed in this work do not represent any undisclosed current or future affiliations.

**Pietro Lesci** received funding from the European Research Council (ERC) under the European Union's Horizon 2020 Research and Innovation programme grant AVeriTeC (Grant agreement No. 865958).

**Max Müller-Eberstein** thanks the IT University of Copenhagen's High-Performance Computing Cluster for supporting the representational stability experiments in §4.

**Naomi Saphra**'s work was enabled in part by a gift from the Chan Zuckerberg Initiative Foundation to establish the Kempner Institute for the Study of Natural and Artificial Intelligence.

**Hailey Schoelkopf** thanks EleutherAI for the opportunity to conduct this work.

**Willem Zuidema** is funded by the Institute for Logic, Language and Computation of the University of Amsterdam.

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

## A    TRAINING DETAILS

We used the `v1.0` version of the GPT-Neox codebase[9] for model training. We report the training loss for the two outlier pre-training runs, 410M seed 3 and 4, below. We refer to the Weights & Biases space for the training losses and logs of all runs.

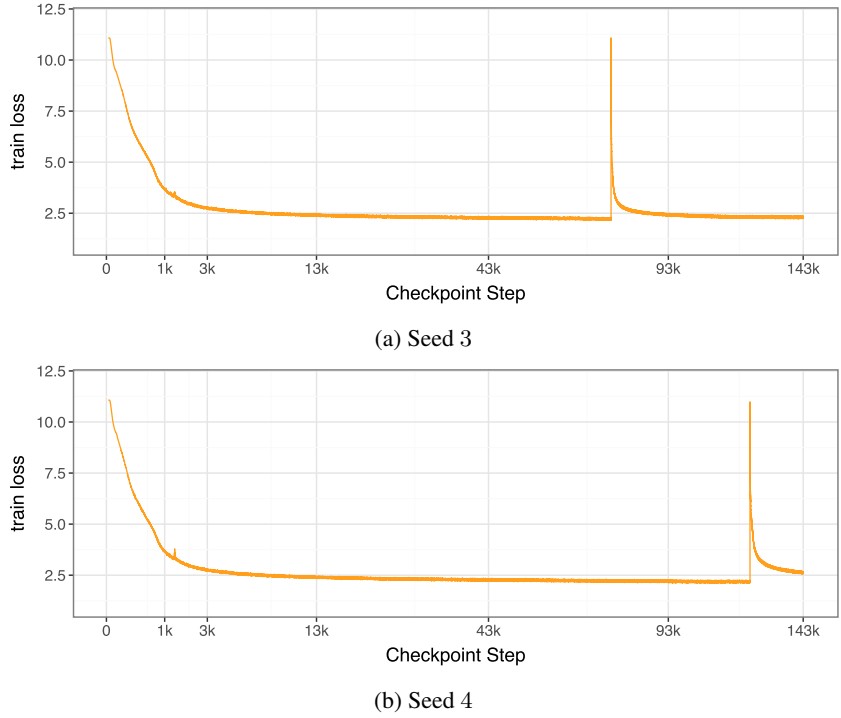

(a) Seed 3

(b) Seed 4

Figure 5: Training loss for 410M outlier seeds 3 and 4 showing "loss spikes".

## B    EVALUATION DATASETS DETAILS

In this section, we describe the datasets used in §3 and §4.

**ARC (Easy) and ARC (Challenge)** (Clark et al., 2018).    The AI2 Reasoning Challenge (ARC) is a benchmark for evaluating natural science question answering. It consists of two subsets, Easy and Challenge, where questions are drawn from standardized science exams. Following Brown et al. (2020), evaluation is performed using log-likelihood-based scoring and reported in terms of accuracy.

**LAMBADA** (Paperno et al., 2016).    LAMBADA is a language modelling benchmark that measures a model's ability to predict the final word of a sentence given its full context. The dataset consists of narrative texts, and successful prediction requires long-range context comprehension.

**Logiqa** (Liu et al., 2020).    Logiqa is a reading comprehension dataset designed to test logical reasoning in language models. The questions are derived from LSAT exam questions and require deductive reasoning and critical thinking skills. Evaluation is performed using multiple-choice accuracy.

**Piqa** (Bisk et al., 2020).    Piqa is a dataset designed to evaluate physical commonsense reasoning. It consists of multiple-choice questions requiring an understanding of everyday physical interactions. Evaluation is based on accuracy.

---

[9]github.com/EleutherAI/gpt-neox tag v1.0.

**SciQ (Welbl et al., 2017).** SciQ is a question-answering dataset focused on scientific topics. It includes multiple-choice, direct-answer, and "cloze"-style questions sourced from educational materials. Evaluation involves measuring accuracy on multiple-choice and direct-answer formats.

**WSC (Levesque et al., 2012).** The Winograd Schema Challenge (WSC) is a coreference resolution task designed to test commonsense reasoning. It consists of sentence pairs that differ by a single word, requiring the model to correctly resolve ambiguous pronouns based on contextual cues. Accuracy is used as the primary evaluation metric.

**WinoGrande (Sakaguchi et al., 2020).** WinoGrande is an expanded version of the Winograd Schema Challenge, containing sentence pairs with minor lexical variations. The task evaluates a model's ability to resolve ambiguous pronouns by comparing the probabilities of different completions. Accuracy is reported as the primary metric.

**BLiMP (Gender Agreement) (Warstadt et al., 2020).** BLiMP (Gender Agreement) is a subset of the BLiMP benchmark designed to evaluate gender bias in language models. It consists of minimal sentence pairs that differ only in gender-marked words, allowing the assessment of whether a model exhibits gender preference in syntactic and morphological structures.

**CrowS-Pairs (Nangia et al., 2020).** CrowS-Pairs assesses biases in language models by presenting minimal sentence pairs that contrast stereotypical and non-stereotypical perspectives on US-protected demographic groups. We use the adaptation by Névéol et al. (2022), which removes instances identified as problematic for validity following Blodgett et al. (2021).

**Simple Co-occurrence Bias (Smith et al., 2022).** Simple Co-occurrence Bias evaluates gender associations in language models by analysing the likelihood of gendered identifiers (e.g., "man", "woman") appearing in simple template-based prompts. Following Brown et al. (2020); Smith et al. (2022), we report directional bias in model predictions. The dataset is available at huggingface.co/datasets/oskarvanderwal/simple-cooccurrence-bias.

**Coref (Pradhan et al., 2013).** Coref is a dataset for coreference resolution, which involves linking mentions of the same entity in a text. We use the coreference annotation layer from OntoNotes 5.0, which is commonly used to evaluate models' ability to resolve pronouns and named entity references in complex passages.

**Dep (Silveira et al., 2014).** Dep is a dataset for dependency parsing, providing annotations of relative syntactic functions based on the English Web Treebank. It is used to evaluate a model's ability to predict grammatical relations between words in a sentence.

**NER (Pradhan et al., 2013).** NER (Named Entity Recognition) is a dataset used to identify and classify proper nouns into 18 predefined categories, such as persons, organizations, and locations. We use the NER annotation layer from OntoNotes 5.0, which serves as a standard benchmark for evaluating entity recognition performance.

**PoS (Pradhan et al., 2013).** PoS (Part-of-Speech Tagging) is a dataset that annotates words in a sentence with one of 51 syntactic categories (e.g., noun, verb, adjective), as taken from OntoNotes 5.0. It evaluates the models' ability to perform syntactic parsing at the word level.

**SemTag (Abzianidze et al., 2017).** SemTag is a dataset for semantic tagging, which classifies words or phrases based on 69 semantic roles and meanings. It is used to assess the models' ability to understand and differentiate word meanings in context.

**Senti (Socher et al., 2013).** Senti is a sentiment analysis dataset containing sentences labelled with sentiment polarity (positive, negative, neutral). It is used to evaluate models' ability to infer sentiment from textual data.

**Topic (Lang, 1995).** Topic is a dataset for topic classification, consisting of documents labelled with 20 predefined topic categories. It serves as a benchmark for evaluating models' ability to perform document-level topic classification.

## C    TRAINING MAPS

In Table 4, we report the metrics used to fit the HMMs to create the training maps in §5. These metrics are (partially) taken from Hu et al. (2023) Appendix B, to which we refer for further details.

| Metric | Description |
|---|---|
| $L_1$ | The $L_1$-norm, averaged over the weight matrices |
| $L_2$ | The $L_2$-norm, averaged over the weight matrices |
| $L_1/L_2$ | Weight sparsity (ratio of their $L_1$ and $L_2$ norms), averaged over the weight matrices |
| $\mu_w$ | Sample mean of the weights |
| $\text{median}_w$ | Median of the weights |
| $\sigma_w$ | Sample variance of weights |
| $\mu_b$ | Sample mean of the biases |
| $\text{median}_b$ | Median of the biases |
| $\sigma_b$ | Sample variance of biases |
| trace | The average trace over the weight matrices |
| $\lambda_{max}$ | The average spectral norm of the weights |
| $\text{trace}/\lambda_{max}$ | The average trace over spectral norm |
| $\mu_\lambda$ | The average singular value over the weights |
| $\sigma_\lambda$ | Sample variance of singular values over the weights |

Table 4: Statistics of model parameters (both weights and biases) used to fit the HMMs in §5.

## D    ADDITIONAL FIGURES

We report the figures for all the benchmarks analysed in §3. The figures follow in the next pages.

## E    RELEASE LINKS

Besides the link to the Hugging Face collection listing all checkpoints reported in the abstract,[10] in Table 5 we report the links to the individual checkpoints on the Hugging Face Hub. Also, the indices used to recreate the pre-shuffled datasets are available at huggingface.co/datasets/EleutherAI/pile-preshuffled-seeds. Links follow on the next page (after the figures).

---

[10]Explicitly, huggingface.co/collections/EleutherAI/polypythias-67bed6916110c8933e1ea561.

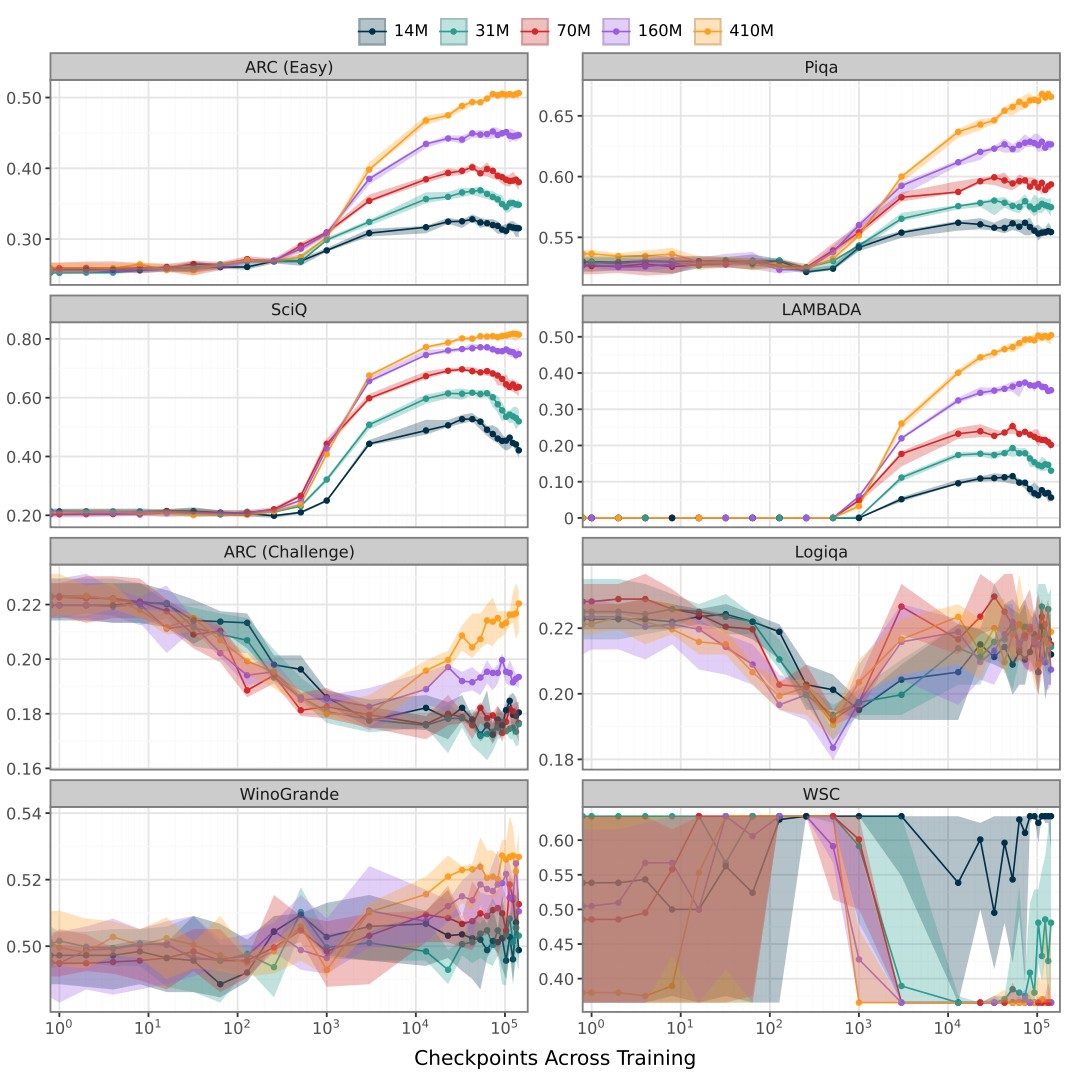

Figure 6: Accuracy (median and interquartile range across seeds) on ARC (Easy), Piqa, SciQ, LAMBADA, ARC (Challenge), Logiqa, WinoGrande, and WSC tasks.

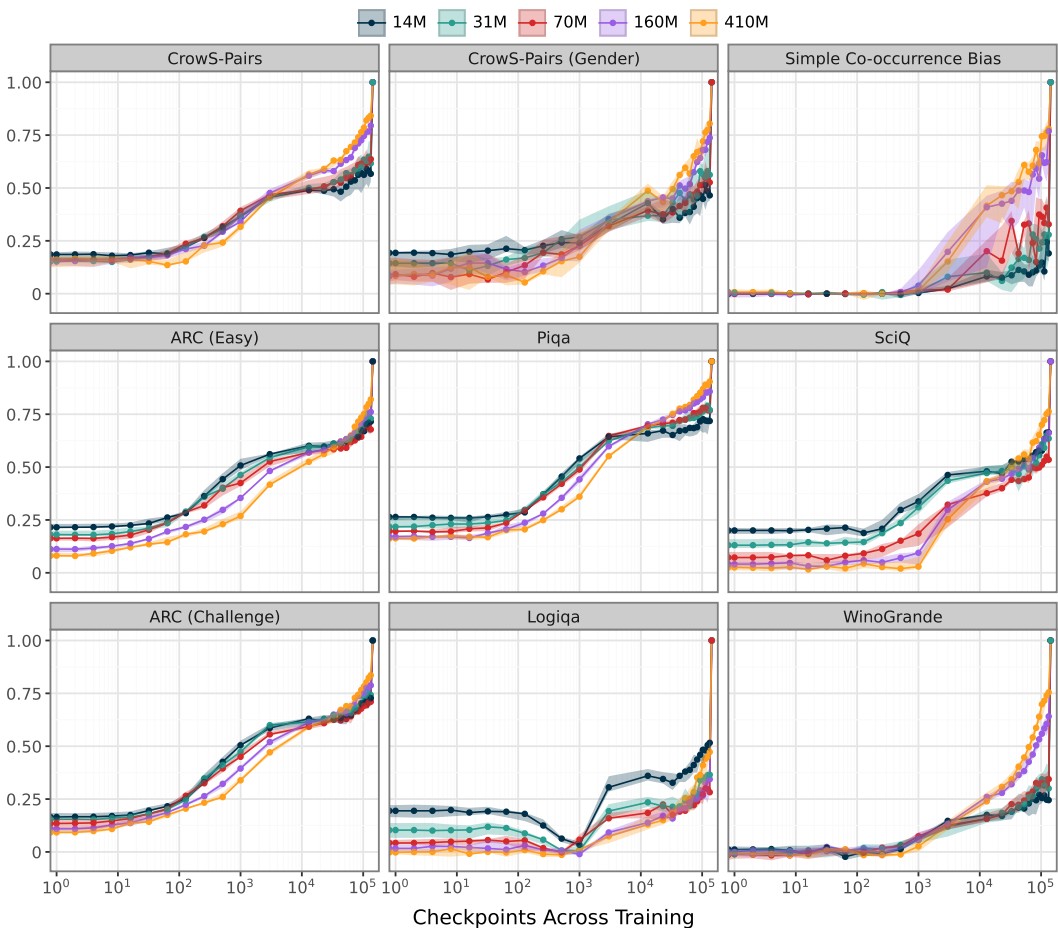

Figure 7: Self-consistency (median and interquartile range across seeds) on CrowS-Pairs, CrowS-Pairs (Gender), Simple Co-occurrence Bias, ARC (Easy), ARC (Challenge), Piqa, SciQ, ARC (Challenge), Logiqa, and WinoGrande.

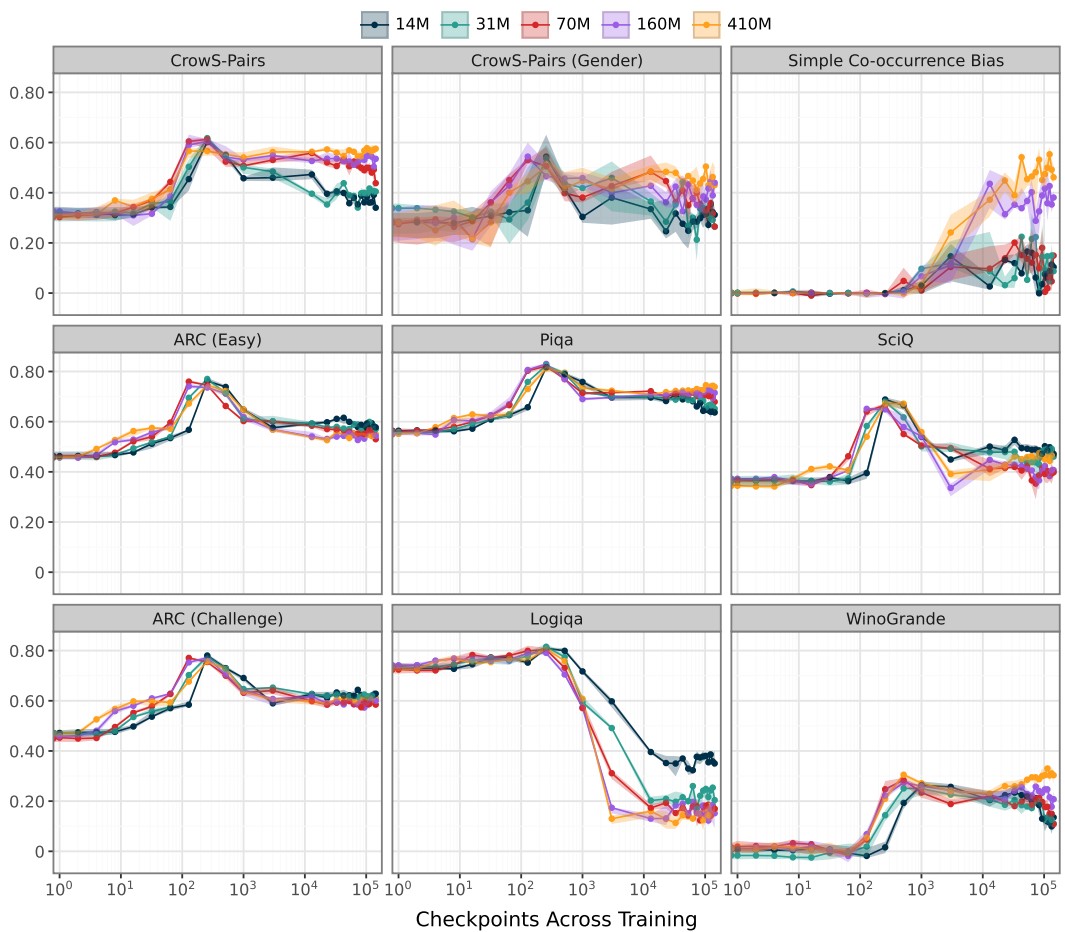

Figure 8: Inter-seed agreement (median and interquartile range across seeds) on CrowS-Pairs, CrowS-Pairs (Gender), Simple Co-occurrence Bias, ARC (Easy), ARC (Challenge), Piqa, SciQ, ARC (Challenge), Logiqa, and WinoGrande.

| Model Size | Seed | Links |
|---|---|---|
| 14M | 0 (default) | huggingface.co/EleutherAI/pythia-14m |
| | 1 | huggingface.co/EleutherAI/pythia-14m-seed1 |
| | 2 | huggingface.co/EleutherAI/pythia-14m-seed2 |
| | 3 | huggingface.co/EleutherAI/pythia-14m-seed3 |
| | 4 | huggingface.co/EleutherAI/pythia-14m-seed4 |
| | 5 | huggingface.co/EleutherAI/pythia-14m-seed5 |
| | 6 | huggingface.co/EleutherAI/pythia-14m-seed6 |
| | 7 | huggingface.co/EleutherAI/pythia-14m-seed7 |
| | 8 | huggingface.co/EleutherAI/pythia-14m-seed8 |
| | 9 | huggingface.co/EleutherAI/pythia-14m-seed9 |
| 31M | 0 (default) | huggingface.co/EleutherAI/pythia-31m |
| | 1 | huggingface.co/EleutherAI/pythia-31m-seed1 |
| | 2 | huggingface.co/EleutherAI/pythia-31m-seed2 |
| | 3 | huggingface.co/EleutherAI/pythia-31m-seed3 |
| | 4 | huggingface.co/EleutherAI/pythia-31m-seed4 |
| | 5 | huggingface.co/EleutherAI/pythia-31m-seed5 |
| | 6 | huggingface.co/EleutherAI/pythia-31m-seed6 |
| | 7 | huggingface.co/EleutherAI/pythia-31m-seed7 |
| | 8 | huggingface.co/EleutherAI/pythia-31m-seed8 |
| | 9 | huggingface.co/EleutherAI/pythia-31m-seed9 |
| 70M | 0 (default) | huggingface.co/EleutherAI/pythia-70m |
| | 1 | huggingface.co/EleutherAI/pythia-70m-seed1 |
| | 2 | huggingface.co/EleutherAI/pythia-70m-seed2 |
| | 3 | huggingface.co/EleutherAI/pythia-70m-seed3 |
| | 4 | huggingface.co/EleutherAI/pythia-70m-seed4 |
| | 5 | huggingface.co/EleutherAI/pythia-70m-seed5 |
| | 6 | huggingface.co/EleutherAI/pythia-70m-seed6 |
| | 7 | huggingface.co/EleutherAI/pythia-70m-seed7 |
| | 8 | huggingface.co/EleutherAI/pythia-70m-seed8 |
| | 9 | huggingface.co/EleutherAI/pythia-70m-seed9 |
| 160M | 0 (default) | huggingface.co/EleutherAI/pythia-160m |
| | 1 | huggingface.co/EleutherAI/pythia-160m-seed1 |
| | 2 | huggingface.co/EleutherAI/pythia-160m-seed2 |
| | 3 | huggingface.co/EleutherAI/pythia-160m-seed3 |
| | 4 | huggingface.co/EleutherAI/pythia-160m-seed4 |
| | 5 | huggingface.co/EleutherAI/pythia-160m-seed5 |
| | 6 | huggingface.co/EleutherAI/pythia-160m-seed6 |
| | 7 | huggingface.co/EleutherAI/pythia-160m-seed7 |
| | 8 | huggingface.co/EleutherAI/pythia-160m-seed8 |
| | 9 | huggingface.co/EleutherAI/pythia-160m-seed9 |
| | 1 (only data) | huggingface.co/EleutherAI/pythia-160m-data-seed1 |
| | 2 (only data) | huggingface.co/EleutherAI/pythia-160m-data-seed2 |
| | 3 (only data) | huggingface.co/EleutherAI/pythia-160m-data-seed3 |
| | 1 (only parameters) | huggingface.co/EleutherAI/pythia-160m-weight-seed1 |
| | 2 (only parameters) | huggingface.co/EleutherAI/pythia-160m-weight-seed2 |
| | 3 (only parameters) | huggingface.co/EleutherAI/pythia-160m-weight-seed3 |
| 410M | 0 (default) | huggingface.co/EleutherAI/pythia-410m |
| | 1 | huggingface.co/EleutherAI/pythia-410m-seed1 |
| | 2 | huggingface.co/EleutherAI/pythia-410m-seed2 |
| | 3 | huggingface.co/EleutherAI/pythia-410m-seed3 |
| | 4 | huggingface.co/EleutherAI/pythia-410m-seed4 |
| | 5 | huggingface.co/EleutherAI/pythia-410m-seed5 |
| | 6 | huggingface.co/EleutherAI/pythia-410m-seed6 |
| | 7 | huggingface.co/EleutherAI/pythia-410m-seed7 |
| | 8 | huggingface.co/EleutherAI/pythia-410m-seed8 |
| | 9 | huggingface.co/EleutherAI/pythia-410m-seed9 |

Table 5: Links to the individual checkpoints in the PolyPythias release.

