# OpenReview forum: "PolyPythias: Stability and Outliers across Fifty Language Model Pre-Training Runs"
_ICLR.cc/2025/Conference — ICLR 2025 Poster_

### Official Review · Reviewer_hNsS · 2024-10-26

**Soundness:** 3
**Presentation:** 3
**Contribution:** 3
**Rating:** 6
**Confidence:** 3

**Summary:**

This work presents PolyPythias, which extends the Pythia model suite with 9 additional training runs (with different seeds) for 5 model sizes (from 14M to 410M). Detailed analyses are performed from different aspects, including downstream performance, intermediate hidden representations and model parameters. Several interesting patterns are discovered with regard to different training phases and identifying outlier seeds. The presented checkpoints as well as the analyses can inspire more future work on the direction of understanding LM training.

**Strengths:**

- Understanding LM training is an important research direction that could lead to better model interpretability and better LM training.
- The analyses and the released checkpoints could help the community to study the stability of LM training.

**Weaknesses:**

- While it is understandable due to limited resources, the model sizes studied in this work is overall small (the largest one is 410M), while commonly utilized LMs are typically larger than 1B.
- Though the analyses are well-presented, there is still a lack of guidances on how we could train better LM models. One potentially interesting thing is to predict outliers to make the training more stable, and it would be nice if there could be more experiments and analyses on this point.

**Questions:**

- For the "inter-seed agreement" metric described in Sec. 3.1, how about comparing all the pairs of models trained with different seeds? Would it lead to similar results compared to only checking against seed 0?
- It would be nice if there could be some highlighted summarizations at the end of each section to describe key takeawys for the analyses.

---

> ### Author Response · Authors · 2024-11-18
> **Response to Review hNsS**
>
> We thank the reviewer for their insightful comments and positive feedback. Below, we address the individual weaknesses (W) and questions (Q).
>
> &nbsp;
>
> **W1. *“\[...\] model sizes studied in this work is overall small \[...\]”***
> We agree that models up to 410M parameters are small compared to many recent models; however, PolyPythias is still valuable for the research community. We repeat for convenience what we wrote in response to `Reviewer yB47` (W1):
>
> > We agree that models up to 410M parameters are small compared to many recent ones. This choice reflects computational constraints, prioritising seed exploration and checkpoint granularity over model size. Despite this limitation, we believe that our suite is still a valuable resource for the research community (as we address in the introduction), as the range of sizes, seeds, and checkpoints allows researchers to study a variety of questions – just as the original Pythia suite continues to do. Those questions include questions about model stability w.r.t. initialisations and data order and, notably, about how similar the behaviour and internal states of smaller, more manageable models are to those of larger models (a key question in, e.g., Mechanistic Interpretability).
>
> &nbsp;
>
> **W2. *“\[...\] predict outliers to make the training more stable \[...\]”***
> While we do not provide specific training guidance, PolyPythias is designed to help researchers explore factors that improve LMs. We agree that predicting outliers is an important question. In Sec. 5, we analyse factors behind unstable runs but find that predicting outliers from early checkpoints remains challenging and beyond this paper’s scope. To support further research, we provide pre-shuffled training tokens for studying data order effects \[1\] and will release six additional seeds with decoupled data-ordering and weight-initialization seeds.
>
> \[1\] Palm: Scaling language modeling with pathways (Chowdhery et al., JMLR 2023\)
>
> &nbsp;
>
> **Q1. *“\[...\] how about comparing all the pairs of models trained with different seeds \[...\]”***
> Comparing all seed pairs would provide additional granularity. Still, we opted for the simpler approach of comparing against seed 0 to align with our goal of highlighting general trends in inter-seed agreement. Our experiments showed consistent trends across methods, whether using (simple) average agreement or Cohen’s kappa. While deeper analyses (e.g., pairwise comparisons, Fleiss’ kappa, or item-specific agreement \[1\]) could be valuable, we hope our work lays the groundwork for such explorations. We will include additional discussion of these considerations in the camera-ready version.
>
> \[1\] Learning Whom to Trust with MACE (Hovy et al., NAACL 2013\)
>
> &nbsp;
>
> **Q2. *“\[...\] summarizations at the end of each section \[...\]”***
> Thanks for the suggestion. We will update the manuscript accordingly.

---

> > ### Comment · Reviewer_hNsS · 2024-11-28
> > **Response**
> >
> > Thank you for the response, which partially addresses some of my concerns. In addition to predicting outliers, it would be nice if there can be more insights and clearer guidelines for LM training, which is still somewhat lacking in the current version. Therefore, I think I will keep my previous score.

---

### Official Review · Reviewer_RBBE · 2024-11-03

**Soundness:** 3
**Presentation:** 3
**Contribution:** 3
**Rating:** 6
**Confidence:** 4

**Summary:**

The paper studies the stability of language model training, specifically focusing on how random seed variation affects downstream performance. The authors release a dataset of 45 additional runs for the Pythia model across various model sizes, of up to 400M. They explore downstream task performance, stability properties and training phase dynamics. Their findings indicate mostly stable training behaviors for the sizes used, with the exception of a few outliers at the larger size.

**Strengths:**

- Impact of the data released: This data will allow other researchers to investigate the questions in this paper and potentially others. Some of the findings, even if unexplained are very interesting: for example the fact that the inter-seed agreement peaks at the moment where models start to leaning meaningful patterns.
- The paper is well written and provides thorough and extensive investigations into the questions addressed. The experimental results in themselves are very interesting and likely to aid in furthering this topic. A better understanding of training dynamics can lead to more efficient and effective use of resources in the community.

**Weaknesses:**

- The representational stability investigation: I am not convinced by the interpretation (or explanation) given to the experiments in this section, namely: 1) It looks like you are building classifiers to solve various tasks based on the hidden representations of the networks (which is a very useful setup in practice for solving those tasks). However the entire setup is cast as a way to investigate the representation stability. Imo this requires more explanation, since it's not stability as e.g. similarity of hidden representations (the most straightforward way to understand it), but stability w.r.t to their ability to solve certain tasks. As such I am struggling to understand the MDL and linear probe interpretation of this setup 2) The sparsity-inducing prior: If random vectors can be used to learn the tasks perfectly then the tasks may not be well designed. If random vectors can be used to learn the task to some extent, in that case just use that as a baseline factor (which I assume is what codelength ratio aims for).

- Gender bias tasks: the authors are not sure if the small benchmark size is causing the results observed. I would recommend making this investigation more thorough and use gender bias data sets released for other tasks. Given that LLMs can be used to address any task, that should not be an issue (I know machine translation has such data sets, but I am sure other tasks as well).

- Interpretability of section 5: motivate why you use feature maps rather than something simpler and more interpretable (see suggestions bellow)

**Questions:**

Questions:
- I find the sparsity inducing prior in equation 1) problematic. Please give more details there: I assume it's kept constant across all setups; or can you design experiments that don't require it (where random representations don't perfectly solve the tasks)?

Suggestions
- in section 4: I would recommend to explain the MDL and linear probe interpretation of this setup a bit better, or alternatively remove it altogether. Specifically, what I see is a setup of training classifiers on top of the hidden representations to solve various tasks. You are then looking at various metrics across different checkpoints: task performance, task performance relative to random baseline, change in classifier parameters. The interpretation that is more natural to me is that you want to investigate the change in model parameters; you are investigating this change in a task-oriented setup, because the relation between parameters when varying the seed is not obvious (although at different training stages that relation can be easily quantified with standard metrics).

- in section 5: it's unclear why you need something as complicated as training maps to explore this. If previous work explained the limitations of using a list of simple interpretable metrics please, do include that here. Table 3 showing the actual features determining transitions is much more illuminating and shows that the data is quite interpretable. I would recommend listing these features at the beginning of section 4 to give the reader a more concrete understanding of this section.

---

> ### Author Response · Authors · 2024-11-18
> **Response to Review RBBE**
>
> We thank the reviewer for their insightful comments and feedback. We address the individual weaknesses (W), questions (Q), and suggestions (S) below.
>
> &nbsp;
>
> We group W1, S1, and Q1 as they refer to the MDL setup.
>
> **W1. *“\[...\] I am not convinced by the interpretation \[...\]”***
> In Sec. 4, we analyse how linguistic information is represented in the latent representations of the model across checkpoints and sizes. The probing framework (e.g., \[5\]) allows us to study this phenomenon by measuring how accurately linguistic information can be decoded from internal representations. Probing is operationalised as a classification task. Thus – as correctly pointed out – our stability results reflect probes' ability to solve certain tasks using models' representations. We will clarify this further in the camera-ready version.
>
> **S1. *“\[...\] I would recommend to explain the MDL \[...\]”***
> Thanks for your suggestion. We justify the MDL setup below eq. 1 in the paper; however, we will revise the manuscript to clarify this point further. Other advantages of MDL are discussed in Q1.
>
> **Q1. *“I find the sparsity inducing prior in equation 1\) problematic \[...\]”***
> The prior is indeed kept constant across all setups: following prior work (\[1, 2\]), we use a zero-mean Jeffreys prior that applies jointly to each probe, with a scaling factor that encourages sparsity \[3, 4\]. We will add a detailed description in the appendix of the camera-ready version.
>
> MDL allows one to distinguish between linguistic information being encoded vs. probe memorisation, thus solving the problem of random vectors being able to learn tasks (a problem that affects standard linear probes trained with cross-entropy; see \[5-6\]). Also, MDL is more robust to hyperparameter choices and enables consistent comparisons between different architectures (\[2\]).
>
> \[1\] Information-Theoretic Probing with Minimum Description Length (Voita & Titov, EMNLP 2020\)
> \[2\] Subspace Chronicles: How Linguistic Information Emerges, Shifts and Interacts during Language Model Training (Müller-Eberstein et al., EMNLP 2023).
> \[3\] Adaptive Sparseness using Jeffreys Prior (Figueiredo, NeurIPS 2001).
> \[4\] Bayesian Compression for Deep Learning (Louizos et al., NeurIPS 2017).
> \[5\] Information-Theoretic Probing for Linguistic Structure (Pimentel et al., ACL 2020).
> \[6\] Language Modeling Teaches You More than Translation Does (Zhang and Bowman, BlackboxNLP 2018).
> \[7\] Designing and Interpreting Probes with Control Tasks (Hewitt and Liang, EMNLP 2019).
> Max
>
> &nbsp;
>
> **W2. *“\[...\] use gender bias data sets released for other tasks. \[,,,\]”***
> We chose to focus on the benchmarks that capture different manifestations of gender (bias) during training: CrowS-Pairs tests semantically more complex gender stereotypes (e.g., “men/women cannot drive” vs. “nurse→she, doctor→he”) while BLiMP and Simple Co-occurrence Bias rely on simple gendered pronoun resolutions. Expanding the analysis to datasets from other tasks (e.g., MT) might provide additional context but it further complicates interpretations as it is unclear whether benchmarks for different tasks measure the same constructs \[1,2\]. Moreover, previous work has identified many potential sources of benchmark “unreliability” \[3,4,5,6\] that require comprehensive analyses (e.g., \[7\]). While valuable, these lie outside this paper's scope, but we will expand our discussion of these issues in the camera-ready version.
>
> \[1\] Language (technology) is power: A critical survey of "bias" in NLP (Blodgett et al., ACL 2020\)
> \[2\] Undesirable biases in NLP: Addressing challenges of measurement (van der Wal et al., JAIR 2024\)
> \[3\] Efficient Benchmarking (of Language Models) (Perlitz et al., NAACL 2024\)
> \[4\] The MultiBERTs: BERT Reproductions for Robustness Analysis (Sellam et al., ICLR 2022\)
> \[5\] Underspecification presents challenges for credibility in modern machine learning (D’Amour et al., JMLR 2022\)
> \[6\] PATCH--Psychometrics-AssisTed benCHmarking of Large Language Models: A Case Study of Mathematics Proficiency (Fang et al., ArXiv 2024\)
> \[7\] fl-IRT-ing with Psychometrics to Improve NLP Bias Measurement (Bachmann et al., Minds and Machines 2024\)
>
> &nbsp;
>
> **W3. *“\[...\] motivate why you use feature maps \[...\]”* \+ S2. *“\[...\] If previous work explained the limitations \[...\]”***
> We will explain the individual features used to construct the training map at the beginning of Sec. 5.
>
> A training map is useful because it allows one to aggregate information from a whole list of low-level features. One could heuristically do this, but training maps are more principled. We understand that the individual metrics (e.g., loss) can provide intuitive information about the training dynamics, but they do not provide a comprehensive picture (and may be a result of the aforementioned low-level features).

---

### Official Review · Reviewer_9Ep5 · 2024-11-04

**Soundness:** 3
**Presentation:** 3
**Contribution:** 3
**Rating:** 6
**Confidence:** 3

**Summary:**

The paper introduces 45 training runs (~7K checkpoints) for 5 model sizes of Pythia, ranging from 14M to 410M parameters across 9 seeds. The work mainly studies model training dynamics and scaling behaviours across seeds and model sizes.

**Strengths:**

- The paper is well-written. The authors support every claim with a thorough experimental setup.

**Weaknesses:**

- I do not find any significant weakness(es). Please refer to other reviews for a collective opinion.

**Questions:**

--

---

> ### Author Response · Authors · 2024-11-18
> **Response to Review 9Ep5**
>
> We thank the reviewer for their positive feedback. We have carefully addressed the comments and suggestions in the other reviews. If our responses to other reviewers alleviate any remaining concerns, we hope the reviewer will consider increasing their rating.

---

### Official Review · Reviewer_yB47 · 2024-11-04

**Soundness:** 3
**Presentation:** 3
**Contribution:** 3
**Rating:** 8
**Confidence:** 2

**Summary:**

The authors created 45 different training runs on 5 existing language model series Pythia 14M to 410M with varying random seeds. The authors named these runs as PolyPythia, and performed several investigation on the model performances and dynamics. Specifically, the authors reported (1) downstream performance and its agreement (between other seeds) and consistency (within the same seed), (2) algebraic analysis on the embedding vectors (through trained probes), and (3) HMM-based transition analysis. Every experiment show that there is a consistent trend among different model sizes and seeds except 2 runs on the largest model they trained. These 2 runs are specifically investigated in detail in the paper, compared with other stable runs.

**Strengths:**

* It is definitely useful if the trained checkpoints in this study are released. Especially, it involves unstable runs (runs with loss spikes), so this will provide clues to understanding loss spikes, which are always a problem when training large language models.
* Probing and HMM analysis are ones of interesting examples of the recent techniques to analyse the inner representation/dynamics of the models.

**Weaknesses:**

* As noted in the paper, the size of the model trained by the authors is limited. Compared with the sizes of recent language models that tend to have usually more than 1B-10B parameters, the size of the maximum model in this paper (410M) is too small and it is still unclear that (1) the same phenomena observed in the paper can be identified in larger models and (2) if other unseen phenomena will happen in the large models.
* It is also still unclear if the phenomena on training are discovered well even in the same range of model sizes that the paper focused on. The paper confirmed its stability of each runs (except the two runs on 410M), it is not revealed that what configuration affects the stability as the experiment only varied the random seed for each model size.

**Questions:**

* It looks the number of checkpoints mapped in every figure is less than that the authors actually captured, especially checkpoints after 10^3. Though I think authors have sampled some of them for some reason, it would be useful to plot all data points.
* As shown in figures, there are remarkable trends between steps 10^2 to 10^4 in all runs. It may be desirable to save and analyse checkpoints at more frequent intervals in this range, rather than following the same schedule in Pythia.
* Related to the second itemizing on the weaknesses, I'm also curious whether we can say that the observed state transisions in the HMM analysis are generic tendency or not, especially for unstable runs, to judge whether we can utilize this tendency to forecast problematic behavior of training runs, or we can use this tool for only analysing the checkpoints *after* they were obtained. It would be better to take more observation around this point, but I also understand that it requires more compute resources.
* In my personal experience on billions-scale LMs, I often observed loss spikes during many runs, but most of them can be recovered very quickly within less than 100 steps, but there are also several spikes that catastrophically destroys the whole model and the recovery requires as many steps as the authors observed in this study. I'm curious if authors observed similar trends to that I noted here or only the two the paper mentioned.

### Errors
* It looks Fig. 1 is strange:
  * It is mentioned with "middle and bottom rows" in p.3, but it looks the correct reference should be "middle and right columns."
  * Some important data points are mentioned in p.4 paragraph "Finding outlier seeds ...", but the figure actually doesn't show them. Especially the paragraph mentioned the two unstable runs, that Fig.1 doesn't include any data points for specific runs.

---

> ### Author Response · Authors · 2024-11-18
> **Response to Reviewer yB47**
>
> We thank the reviewer for their insightful comments and positive feedback. Below, we address the individual weaknesses (W) and questions (Q).
>
> &nbsp;
>
> **W1. *“As noted in the paper, the size of the model trained by the authors is limited \[...\]”***
> We agree that models up to 410M parameters are small compared to many recent ones. This choice reflects computational constraints, prioritising seed exploration and checkpoint granularity over model size. Despite this limitation, we believe that our suite is still a valuable resource for the research community (as we address in the introduction), as the range of sizes, seeds, and checkpoints allows researchers to study a variety of questions – just as the original Pythia suite continues to do. Those questions include questions about model stability w.r.t. initialisations and data order and, notably, about how similar the behaviour and internal states of smaller, more manageable models are to those of larger models (a key question in, e.g., Mechanistic Interpretability).
>
> &nbsp;
>
> **W2. *“It is also still unclear if the phenomena \[...\]” \+* **Q3.** *“\[...\] observed state transisions in the HMM analysis are generic tendency or not \[...\]***
> You are correct: PolyPythias capture (by chance) training instabilities (i.e., “loss spikes”) for two of the 50 runs considered. In fact, we proactively tried to minimise unstable runs to obtain as many well-trained models as possible. Importantly, in creating the multi-seed collection, we want to allow researchers to study a broader definition of stability (beyond “loss spikes”). For example, how stable are the learned representations across model size and seed? Or, how consistent are the mechanisms/circuits that emerge during training? How reliable are existing benchmarks w.r.t. the data order/model seed? Answers to these questions are orthogonal to training instabilities as signified by “loss spikes”. Finally, we note that understanding the cause of “loss spikes” would require experiments specifically designed for this purpose (e.g., \[1,2\]), which are outside the scope of this paper. We will make this clearer in the next version.
>
> \[1\] Latent State Models of Training Dynamics (Hu et al., TMLR 2024\)
> \[2\] Stabilizing transformer training by preventing attention entropy collapse (Zhai et al., PMLR, 2023\)
>
> &nbsp;
>
> **Q1. *“\[...\] it would be useful to plot all data points.”***
> To limit computational costs, for each size and seed, we evaluate performance on a subset of the available checkpoints–(log-spaced) steps 0,1,2,...,512, 1k, and from step 3k onwards we choose every 10k-th step up to 143k (Sec 3.1). For the 14M model, we ran initial probing experiments for all checkpoints. While not representative of all setups, the resulting trajectories were highly consistent with what we observed for the final subsampled checkpoints. We hope this clarifies the questions.
>
> &nbsp;
>
> **Q2. *“\[...\] save and analyse checkpoints at more frequent intervals \[...\]“***
> We do agree that more checkpoints would be useful. However, given the size of this release (i.e., 7000 checkpoints) we kept the same checkpointing frequency as the original Pythia suite. This rate is consistent or higher than prior efforts for publishing intermediate checkpoints with comprehensive coverage (e.g., MultiBERTs \[1\], BLOOM \[2\], Mistral \[3\]).
>
> \[1\] The MultiBERTs: BERT Reproductions for Robustness Analysis (Sellam et al., ICLR 2022\)
> \[2\] BLOOM: A 176B-Parameter Open-Access Multilingual Language Model (Scao et al., BigScience Workshop 2022\)
> \[3\] [https://github.com/stanford-crfm/mistral](https://github.com/stanford-crfm/mistral) (Mistral, 2022\)
>
> &nbsp;
>
> **Q4. *“\[...\] I'm curious if authors observed similar trends to that I noted here or only the two the paper mentioned.”***
> This is indeed an interesting question. In our analysis (Fig 4), we observe that runs that diverge try to recover by returning to a previous training state. Due to space constraints, we have not investigated this further in the paper. However, addressing this type of question is exactly the goal of PolyPythias. We will extend the discussion of this point in the camera-ready version appendix.

---

> > ### Comment · Reviewer_yB47 · 2024-12-03
> >
> > Thanks for providing detailed answers to my question. As the rebuttals sound acceptable and I already rated this paper accepted, I'd keep my recommendation as is.

---

### Meta-Review · Area_Chair_b6bM · 2024-12-19

**Metareview:**

The paper introduces 45 additional training runs for the Pythia model suite across five sizes (14M to 410M parameters) and nine random seeds, producing ~7000 checkpoints. The work investigates training stability, downstream performance consistency, and the emergence of training phases, revealing that training behavior is largely stable across seeds, with notable exceptions in two outlier runs of the largest model. The release of these checkpoints provides a valuable resource for the community to study training dynamics, stability, and interpretability.

The paper's strengths lie in its rigorous analysis and dataset contribution, which will enable further exploration of training instabilities and scaling trends. While the largest models studied (410M) are small relative to current state-of-the-art LMs, the findings remain relevant, particularly for mechanistic interpretability. Reviewers noted the paper’s clarity, methodology, and potential for enabling future research, though they noted the absence of predictive tools for identifying outliers and practical training guidance as minor limitations. The authors adequately addressed concerns regarding resource constraints and methodology during the rebuttal.

The primary weaknesses of the paper are tied to the relatively small models sizes (max 410M parameters) compared to modern large-scale language models, raising questions about the generalizability of the findings to models exceeding 1B parameters. Additionally, the paper stops short of providing actionable insights or predictive tools for identifying or mitigating outliers during training, which could improve practical LM training strategies. Some reviewers also raised concerns about the interpretation of the probing setup used to measure representation stability, suggesting it could be clarified or complemented with alternative methods. Lastly, the checkpoint sampling frequency, particularly in critical training intervals, was noted as a limitation that might obscure finer-grained training dynamics. While these weaknesses are valid, they do not undermine the paper's overall contribution.

Overall, the paper provides a significant resource and meaningful insights into LM training stability, outweighing its limitations. I recommend accepting the paper due to its strong contribution to a critical area of research and its value as a foundation for future work on larger models and predictive stabilization techniques.

**Additional Comments On Reviewer Discussion:**

During the rebuttal period, the main concerns raised by reviewers focused on three key areas: the generalizability of findings to larger models, the practical impact of the analysis, and the clarity of certain experimental setups.

Model Size and Generalizability: Reviewers, particularly yB47 and hNsS, noted that the largest model studied (410M parameters) is small compared to modern large-scale LMs. They questioned whether the observed stability trends and findings would generalize to models exceeding 1B parameters. The authors acknowledged this limitation but justified their focus on smaller models due to computational constraints, emphasizing that the findings remain relevant for understanding training dynamics and serve as a foundation for further research. While this concern is valid, the reviewers agreed that the paper’s insights are nevertheless valuable.

Practical Impact and Predictive Tools: Reviewers, especially hNsS, highlighted the absence of actionable tools or clear guidelines for improving training stability, such as predicting outlier runs. The authors responded by clarifying that their primary goal was to enable further research by releasing a comprehensive dataset and performing analyses to identify training patterns.

Clarity of Analysis: Reviewer RBBE raised concerns about the interpretation of the probing experiments for measuring representation stability, specifically the reliance on task performance rather than direct metrics of representation similarity. The authors clarified their use of Minimum Description Length (MDL) to distinguish between encoded information and probe memorization, committing to improving the explanation in the camera-ready version. While some reservations remained, the clarification addressed most of the concerns. Additionally, minor issues like checkpoint sampling frequency and the completeness of certain visualizations were discussed, with the authors explaining their methodology and computational trade-offs.

Overall, the rebuttal period clarified key points. While concerns about generalizability and practical impact remain, the reviewers agreed that the paper’s contributions—particularly the release of a large, multi-seed checkpoint dataset—provide significant value to the community. Weighed against its limitations, the paper’s strengths justify its acceptance, as it lays a solid foundation for further research on training stability and interpretability in language models.

---

### Decision · Program_Chairs · 2025-01-22

Accept (Poster)